Self-regulatory mode (locomotion and assessment), well-being (subjective and psychological), and exercise behavior (frequency and intensity) in relation to high school pupils’ academic achievement

Garcia Danilo 1 2 danilo.garcia@icloud.com
Jimmefors Alexander 2 3
Mousavi Fariba 2
Adrianson Lillemor 2 4
Rosenberg Patricia 2
Archer Trevor 2 3
1 Institute of Neuroscience and Physiology, Centre for Ethics, Law and Mental Health, University of Gothenburg , Gothenburg , Sweden
2 Network for Empowerment and Well-Being , Sweden
3 Department of Psychology, University of Gothenburg , Gothenburg , Sweden
4 The Academy of Library, Information, Pedagogy and Information Technology, University of Borås , Borås , Sweden
Abdullah Jafri
Electronic publication date: 2015 Apr 2
Publication date: 2015
Volume: 3
Electronic Location ID: e847
Received 2014 Sep 3; Accepted 2015 Mar 2
Copyright: © 2015 Garcia et al.
Copyright year: 2015
Copyright holder: Garcia et al.
License: This is an open access article distributed under the terms of the Creative Commons Attribution License, which permits unrestricted use, distribution, reproduction and adaptation in any medium and for any purpose provided that it is properly attributed. For attribution, the original author(s), title, publication source (PeerJ) and either DOI or URL of the article must be cited.
License URL: https://creativecommons.org/licenses/by/4.0/

Keywords: Academic achievement, Assessment, Psychological Well-Being, Grades, Self-regulation, Locomotion

Funding: Stiftelsen Kempe-Carlgrenska Fonden AFA Insurance Dnr. 130345 FORTE Dnr. 2013-2923 This research was supported by the Stiftelsen Kempe-Carlgrenska Fonden, AFA Insurance (Dnr. 130345), and FORTE (Dnr. 2013-2923). The funders had no role in study design, data collection and analysis, decision to publish, or preparation of the manuscript.

==============================
Background. Self-regulation is the procedure implemented by an individual striving to reach a goal and consists of two inter-related strategies: assessment and locomotion. Moreover, both subjective and psychological well-being along exercise behaviour might also play a role on adolescents academic achievement.

Method. Participants were 160 Swedish high school pupils (111 boys and 49 girls) with an age mean of 17.74 (sd = 1.29). We used the Regulatory Mode Questionnaire to measure self-regulation strategies (i.e., locomotion and assessment). Well-being was measured using Ryff’s Psychological Well-Being Scales short version, the Temporal Satisfaction with Life Scale, and the Positive Affect and Negative Affect Schedule. Exercise behaviour was self-reported using questions pertaining to frequency and intensity of exercise compliance. Academic achievement was operationalized through the pupils’ mean value of final grades in Swedish, Mathematics, English, and Physical Education. Both correlation and regressions analyses were conducted.

Results. Academic achievement was positively related to assessment, well-being, and frequent/intensive exercise behaviour. Assessment was, however, negatively related to well-being. Locomotion on the other hand was positively associated to well-being and also to exercise behaviour.

Conclusions. The results suggest a dual (in)direct model to increase pupils’ academic achievement and well-being—assessment being directly related to higher academic achievement, while locomotion is related to frequently exercising and well-being, which in turn, increase academic achievement.

From a social perspective, education exerts a strong influence on both the individual and society since it raises the standard of living (Kessler et al., 1995). On an individual level, education promotes the sense of personal control (e.g., one aspect of an individual’s psychological well-being), a healthy lifestyle, greater income, employment, interpersonal relations, and social support (Mirowsky & Ross, 2003). Hence, it is important that individuals who work with any form of education (for example, teachers and other personnel in the educational system) have knowledge of the different factors that might help students’ to achieve both academic goals and high levels of well-being. Self-regulatory theory (Kruglanski et al., 2000), for instance, suggest that individuals implement two modes or strategies when they strive to reach a goal: (1) assessment, which refers to the initial part of the self-regulation process, that is, when the individual evaluate possible goals and procedures needed to attain specific goals and (2) locomotion, which refers to the action-based part of self-regulation when the individual takes action and adheres to a step-by-step procedure until the goal is reached.

Unlike Rubicon’s model of action phases (Gollwitzer, 1990; Heckhausen, 2000), Kruglanski et al. (2000) and Higgins, Kruglanski & Pierro (2003) have studied locomotion and assessment separately and conceptualized them in accordance to Lewin’s distinction between setting a goal and striving to achieve the goal (Lewin et al., 1944). In other words, assessment and locomotion are two separate and individual personality orientations that may vary between and within individuals (Higgins, Kruglanski & Pierro, 2003; Kruglanski et al., 2000). If a person is high in the assessment orientation, she/he tends to rigorously reflect and evaluate different possible pathways and goals. This type of person often evaluates both her/his personality and behaviour. In contrast, individuals who are high in locomotion focus primarily on achieving goals and moving forward. These individuals reflect and evaluate possible pathways and goals very briefly and get going with the action part of the task, or as accurately depicted by the Nike slogan: “they just do it” (Kruglanski et al., 2000).

Assessment is positively correlated with depression and anxiety and negatively correlated with self-confidence and optimism, while locomotion is positively correlated with self-confidence and optimism but negatively correlated with anxiety and depression (Kruglanski et al., 2000; Higgins, Kruglanski & Pierro, 2003). Moreover, there is a positive relationship between locomotion and extrovert behavior (e.g., the ability to socialize with others, positive affectivity), motivation, Type–A behavior, readiness for action, and vitality (Higgins, 1997; Kruglanski et al., 2000), and higher affective well-being (Giacomantonio, Mannetti & Pierro, 2013). In other words, the tendency to constantly evaluate oneself, which is typical for the assessment-oriented person, prompts a sense of inadequacy, negative emotions, lower self-esteem and less optimism (Kruglanski et al., 2007; Jimmefors et al., 2014). Locomotion, on the other hand, implying activation, proactivity, and forward-striving, is linked with lesser reflection and more goal-oriented movement, thereby providing more positive emotions, higher optimism and self-confidence (Kruglanski et al., 2007; Pierro et al., 2013; Jimmefors et al., 2014). This suggests that the two self-regulatory modes are differently related to well-being.

Modern research suggests two definitions of well-being: subjective well-being (Diener, 1984) and psychological well-being (Ryff, 1989). Subjective well-being, or happiness, consists of a cognitive and an affective component (Schimmack, 2007). The cognitive component consists of individuals’ degree of life satisfaction and the affective component of individuals’ experiences of positive and negative feelings (Diener, 1984). Thus, a happy adolescent is satisfied with her/his life and experiences greater levels of positive emotions than negative emotions (Diener, 1984). Psychological well-being is a multi-faceted concept composed of six different factors that, when measured together, would provide an index of an individual’s psychological well-being: self-acceptance, positive relationships with others, autonomy, environmental mastery, purpose in life and personal growth (Ryff, 1989; Ryff, 1995). In this context, the locomotion orientation not only recalls of the “just do it” slogan, but is probably related to the individual’s exercise behavior. Good exercise habits have, in turn, been shown to be positively related to high levels of well-being (Silvestri, 1997; Lotan, Merrick & Carmeli, 2005a; Lotan, Merrick & Carmeli, 2005b).

For instance, good exercise habits (e.g., frequently engaging in intensive physical training) during adolescence provide a more positive and healthy lifestyle, which in turn provide a good prognosis for early prevention of many chronic diseases that originate in early childhood (Twisk, 2001; Trost, 2009). Other studies have confirmed that adolescents who report frequent physical activity also report less stress and depression (Norris, Carroll & Cochrane, 1992). Moreover, physical exercise seem to exert effects at both physical and mental health levels, effects that positively influence adolescents’ academic performance (Dwyer et al., 2001). Thus, adolescents who exercise frequently and with a high level of effort might be able to influence not only their well-being but their academic achievement as well.

Factors influencing academic achievement

Kruglanski et al. (2000) have tested how self-regulation influences academic achievement among 665 high school pupils and found that the individuals who were high in both assessment and locomotion had the highest score in their grade point average. In this context, earlier pedagogical research has suggested self-regulated learning as the method to enhance academic achievement (Corno & Mandinach, 1983; Corno & Rohrkemper, 1985; Pintrich & de Groot, 1990). Self-regulated learning is based on meta-cognition (e.g., thinking about one’s thinking), strategic action (e.g., goal-planning, goal-monitoring, self-evaluation, modification of one’s thoughts and cognitions, as well as evaluating different ways to complete a goal), and pupils’ motivation to learn (Pintrich, 2004). In other words, self-regulated learning includes the assessment mode of self-regulatory theory, but does not include the locomotion mode. This is important, because locomotion is positively associated to well-being, while assessment alone is related to low well-being (Kruglanski et al., 2000).

Subjective well-being has, for example, been proved to be positively related to university students’ final psychology grades (Borrello, 2005), Lebanese university students’ grade point average (Ayyash-Abdo & Sánchez-Ruiz, 2012), and middle-school pupils’ grades (Quinn & Duckworth, 2007; Huebner & Gilman, 2003). High self-efficacy, a person’s belief that she/he has the capacity to achieve a goal or a specific outcome (Bandura, 1977; Bandura, 1982), is related to higher grades and better results on different tasks in school, seatwork, exams, quizzes, essays, reports and higher grades (Pintrich & de Groot, 1990). Self-efficacy is positively related to psychological well-being (Ryff, 1989), which in turn is positively related to positive affect (Garcia, 2011; Garcia, Nima & Kjell, 2014; Garcia et al., 2012). Although these inferences can be drawn, the research linking psychological well-being and academic achievement is rather scarce.

As earlier mentioned, physical exercise exerts a positive effect brain stimulation and enhances learning (Brink, 1995). Moreover, Archer & Garcia (2014a) reviewed research on this association and concluded that both the amount of time, frequency, and the intensity of exercise, are related to high school pupils academic achievement and both physical and mental health. Even acute aerobic exercise interventions (12 min long), seem to improve selective attention and reading comprehension among adolescents (Tine, 2014), at least among those coming from low-income households. In the Swedish context, schools schedule less physical education than ever before (Ericsson, 2005) despite the fact that children who participate in physical activity display better motor fitness, better academic performance and attitude toward schools compared to their sedentary peers (Trost, 2009; Myers et al., 1996). Finally, although the relationship between locomotion and exercise habits seems obvious, to the best of our knowledge no other study has investigated if high locomotion is associated to frequently engaging in intensive physical training (see Archer & Garcia, 2014a; Archer & Garcia, 2014b, for an editorial in which part of this data is presented).

The present study investigates if self-regulation, well-being, and exercise behavior exert essential roles in influencing academic achievement among Swedish high school pupils.

Method

Ethical statement

The research protocol (e.g., confidentiality, participants’ data will not be used for commercial or other non-scientific purposes) was approved by the University of Gothenburg and required only informed consent from participants.

Participants and procedure

The participants were 160 Swedish high school pupils (111 boys and 49 girls) with an age mean of 17.74 (SD = 1.29) from which we obtained 156 valid responses (response ratio = 98%). The pupils attended two different schools on the Swedish west coast. The first school had five hundred pupils and is situated in Gothenburg. The second school is located in a smaller city with eight hundred pupils. Due to the different geographic locations, the pupils social backgrounds are varied. The principals were informed about the study and they confirmed the retrieving of the grades. The pupils were selected by the criteria of being in their last year of high school in order to have final grades in the targeted subjects. The tests were completed online using a link that was sent by email to the respondents. The session took place during the students original classes, and it took about forty minutes to complete the test. Pupils were briefly informed about the test (e.g., confidentiality, right to drop out at any time, etcetera) and asked for their consent to participate. In the first school, everyone who participated received a cinema ticket, while in the second school the participants took part in a lottery for the remaining tickets. The pupils were asked to provide their social security number in order to match their answers with their final grades. The grades were provided by administrative personnel at the schools and were sent to the research team.

Measures

Self-regulation

The Swedish version (Jimmefors et al., 2014) of the Regulatory Mode questionnaire (Kruglanski et al., 2000) was used to assess self-regulatory mode/orientations. The test consists of 30 items measured on a 6 point likert scale (from 1 = strongly disagree to 6 = strongly agree) measuring assessment (e.g., “I spend a great deal of time taking inventory of my positive and negative characteristics,” “I am a critical person”) and locomotion (“I am a doer,” “When I get started on something, I usually persevere until I finish it”). The Cronbach’s α in the present studie were .74 for locomotion and .75 for assessment.

Subjective well-being

The Positive Affect Negative Affect Schedule (Watson, Clark & Tellegen, 1988) assesses the affective component of subjective well-being by requiring participants to indicate on 5-point Likert scale (1 = very slightly, 5 = extremely) to what extent they generally experienced 10 positive (e.g., interested, proud, and strong) and 10 negative emotions (e.g., afraid, nervous, guilty) in the last few weeks. The Temporal Satisfaction With Life Scale (Pavot, Diener & Suh, 1998) assess the cognitive component of subjective well-being and comprises 15-items (7-point likert scale; 1 = strongly disagree, 7 = strongly agree) organized in three subscales assessing past (e.g., “If I had my past to live over, I would change nothing”), present (e.g., “I would change nothing about my current life”), and future life satisfaction (e.g., “There will be nothing that I will want to change about my future”). These instruments showed the following Cronbach’s α in the present study: .85 for positive affect, .85 for negative affect, and .94 for the whole Temporal Satisfaction with Life Scale. The Swedish versions of both instruments have been used in previous studies among adolescents (e.g., Garcia & Erlandsson, 2011; Nima, Archer & Garcia, 2012; Nima, Archer & Garcia, 2013; Schütz, Archer & Garcia, 2013; Garcia, Rosenberg & Siddiqui, 2011). In the present study we used the standardized scores (z-scores) of the temporal life satisfaction total score (i.e., the sum of the past, present, and future subscales) and the difference between participants’ positive affect and negative affect to create the subjective well-being score (i.e., ztemporal life satisfaction + zpositive affect – znegative affect). This procedure has been earlier used in other studies (e.g., Sheldon & Elliot, 1999; Sheldon & Kasser, 1998; Sheldon & Lyubomirsky, 2006; Sheldon et al., 2010; Garcia & Moradi, 2012).

Psychological well-being

The Swedish version (Garcia & Siddiqui, 2009; Garcia, 2012) of Ryff’s short test (Clarke et al., 2001) was used to operationalize psychological well-being. The instrument consists of 18 items with a 6 point Likert scale (from 1 = strongly disagree to 6 = strongly agree) measuring the six dimensions of psychological well-being: autonomy (e.g., “I have confidence in my opinions, even if they are contrary to the general consensus”), environmental mastery (e.g., “In general, I feel I am in charge of the situation in which I live”), personal growth (e.g., “I think it is important to have new experiences that challenge how you think about yourself and the world”), positive relations with others (e.g., “People would describe me as a giving person, willing to share my time with others”), purpose in life (e.g., “Some people wander aimlessly through life, but I am not one of them”), and self-acceptance (e.g., “I like most aspects of my personality”). The Swedish version has showed low reliability for many of the subscales (e.g., Garcia & Siddiqui, 2009), therefore the total score for psychological well-being was used in the present study which showed a Cronbach’s α of. 79.

Exercise behavior

The background questionnaire included two items to measure frequency (“How often do you exercise?”: 1 = never, 5 = 5 times/week or more) and intensity (“Estimate the level of effort when you exercise”: 1 = non or very low, 10 = Very High) of exercise behavior (Garcia et al., 2012). The participants’ answers to both exercise-items (r = .50) were first standardized (i.e., transformed to z-scores) in order to summarize them into a composite measure for exercise behavior; that is, the Archer-Garcia Ratio (Garcia & Archer, 2014a; Garcia & Archer, 2014b). A principal components analysis, with oblimin rotation, suggested that a single primary factor accounted for at least 70.94% of the variance, thus supporting the calculation of the Archer-Garcia Ratio.

Academic achievement

This variable was operationalized through pupils’ final grades in Swedish, Mathematics, English, and Physical Education. The courses take place during either one or two semesters and the grading scale ranges from A = pass with distinction to F = fail. The grades where transformed to “points” accordingly to the Swedish National Agency for Education: A = 20, B = 17.5, C = 15, D = 12.5, E = 10, F = 0, - = − 10 (http://www.studera.nu/download/18.4149f55713bbd91756380003453/gymnasietgy2011.pdf). A grade point average was then computed by simply summarizing the points for each subject and then divided by the number of subjects (i.e., four). Cronbach’s α for the grade point average was .75.

Statistical treatment

For the statistical analyses, a Pearson correlation analysis was conducted to identify the expected correlations between self-regulation (locomotion and assessment), well-being (subjective and psychological well-being), exercise behavior (The Archer-Garcia Ratio), and academic achievement (i.e., grade point average). A multiple regression analysis using the enter method was also conducted in order to investigate this relationship further. See Table 1 for results of Pearson’s correlation. As recommended by Ferguson (2009), we focused on correlations (r) equal or larger than .20 as a minimum effect size presenting a “practically” significant effect for social science data (for r2 the minimum recommended is .04).

Table 1 Correlations among locomotion, assessment, subjective well-being, psychological well-being, Archer–Garcia ratio and grade point average.

	1	2	3	4	5	6	
(1) Locomotion	–						
(2) Assessment	.09	–					
(3) Subjective well-being	.48***	−.23**	–				
(4) Psychological well-being	.59***	−.15	.77***	–			
(5) The Archer-Garcia ratio	.20*	.11	.24**	.14	–		
(6) Grade Point Average	.17**	.25**	.31***	.22**	.23**	–	
Mean and Sd.	3.81±.66	3.76±.71	−.01± 1.79	4.19±.64	10.47±2.82	14.51±2.57	
Cronbach’s α	.74	.75	–	.79	–	.75	
Notes.

The colors highlight correlations higher than .20: Black, Grade Point Average; Blue, Assessment; Yellow, Locomotion; Green, Well-Being.

* p < .05

** p < .01

*** p < .001

Table 2 The regression analysis that shows the influences of locomotion, assessment, subjective well-being, psychological well-being and Archer-Garcia ratio on grade point average.

Predictor variable	Outcome variable	Unst. B	Unst. SE	Stand. β	t	
Locomotion	Grade point average	−.28	.37	−.08	−.75	
Assessment	.72	.32	.21	2.24*	
Subjective well-being	.41	.18	.32	2.25*	
Psychological well-being	.21	.54	.06	.38	
The Archer-Garcia ratio		.10	.08	.11	1.95	
Notes.

The colors correspond to those in Table 1 and Fig. 1: Black, Grade Point Average; Blue, Assessment; Yellow, Locomotion; Green, Well-Being.

* p < .05

Adj. R2 =.11, F = 3.76, p < .001

Results

As expected, academic achievement was positively related to assessment (r = .25, p < .01; r2 = .06). The results also showed that academic achievement was positively related to both subjective well-being (r = .31, p < .01; r2 = .09) and psychological well-being (r = .22, p < .01; r2 = .05). Also as expected, locomotion was positively related to subjective well-being (r = .48, p < .01; r2 = .23) and psychological well-being (r = .58, p < .01; r2 = .33), while assessment on the other hand was negatively related to subjective well-being (r = − .23, p < .01; r2 = .05). Although we expected both measures of well-being to be related to the Archer-Garcia Ratio, only subjective well-being was positively related to exercise behavior (r = .24, p < .01). Nevertheless, the Archer-Garcia Ratio was positively related to academic achievement (r = .23, p < .01; r2 = .06). Finally, as expected, locomotion was positively associated to the Archer-Garcia Ratio (r = .20, p < .05; r2 = .04). Further analysis using multiple linear regressions showed that assessment (β = .21, p < 0.001) and subjective well-being (β = .31, p < 0.001) had an significant effect on grade point average (Table 2). Hence, this suggested that, when controlling for all variables in the study, the main predictors of higher final grades were assessment and subjective well-being.

Discussion

The purpose of this study was to investigate how high school pupils’ self-regulated orientation, well-being and exercise habits are related to academic achievement. Assessment was positively correlated with grade point average, while locomotion was weakly associated to grade point average (see Ferguson, 2009, who suggest .20 as the minimum effect size presenting a “practically” significant effect for social science data). This result was expected because of assessment’s similarity to self-regulated learning—assessment involves strategic thinking, assessing different goals and pathways to achieve these goals (Kruglanski et al., 2000), while self-regulated learning emphasizes assessing one’s own cognitions and thinking and also elaboration and constant renewal and development of one’s learning strategies (Pintrich, 2004). Indeed, previous research has shown that self-regulated learning (e.g., pupils who have developed cognitive strategies in order to plan, monitor and modify their cognitive functions) predicts higher grades in school (Pintrich & de Groot, 1990; Corno & Mandinach, 1983; Corno & Rohrkemper, 1985). Other studies show that locomotion orientation is positively related to Type-A behavior expressed by impatience, competitiveness and a ‘winner-mentality’ (Perry et al., 1990); thus, it is plausible to suggest that high levels of locomotion might interfere with pupils ability to study and perform well in typical school work. Perhaps explaining the weak correlation between locomotion and grade point average in the present study.

Moreover, grade point average correlated positively with subjective well-being and psychological well-being that is in line with Batenburg-Eddes & Jolles’s (2013) findings that concluded that emotional well-being was associated with school children’s underachievement. High subjective well-being is, indeed, defined by high levels of positive affect, low levels of negative affect, and high satisfaction with life (Schimmack, 2007). On the other hand, psychological well-being consisted of several constructs such as self-acceptance, autonomy, tolerance towards others, goal-directed behavior, and self-efficacy (i.e., agentic or self-directed behavior). Agency or the tendency of being proactive, persistent, goal-oriented, for instance, is part of a mature character (Cloninger, 2004) whereby mature individuals with positive self-attitude are more likely to achieve higher grades and happiness (Kjell et al., 2013). While assessment was positively correlated with grade point average, it was negatively correlated with subjective well-being. Possibly, high achieving pupils experience pressures, from parent and significant others, towards achievement that might lower affective status and subjective well-being. Also, ruminating about one’s performance and choices might lead to unhappiness (Nima, Archer & Garcia, 2012; Nima, Archer & Garcia, 2013). This suggests that, while assessment is good for high grades, it might be harmful for pupils’ own happiness. In contrast, locomotion was positively related to both subjective and psychological well-being. This is important because it implies that seeing self-regulation as a dual construct (e.g., Kruglanski et al., 2000) suggested that schools need to promote self-regulating learning for good grades and locomotion for well-being, which in turn might also lead to higher grades.

Along this line, exercise behavior was positively associated to locomotion, subjective well-being and higher grades. Earlier research suggests that physical education increases academic achievement (e.g., Carlson et al., 2008) and that pupils who enjoy physical exercise and sports outside school might also achieve a higher level of physical fitness, which correlates with higher grades (e.g., Johnson, 2008). In other words, exercise interventions may offer an important factor for pupils striving for better grades within several domains, affective, cognitive, health, and with additional effects on the learning process, if not the grade-level. Recent studies (e.g., Tine, 2014) show that even acute aerobic exercise interventions (12-min long) might improve learning. Together with our results, we suggest that exercise might increase grades because it increases well-being.

Limitations and future studies

One limitation in the present study is the measure for exercise behavior, which was self-reported. Nevertheless, in a population of 158 participants at a training facility, the Archer–Garcia Ratio predicted how often individuals had trained for the past six months, measured electronically, even when compared to larger validated scales of exercise behaviour. Thus, the Archer-Garcia Ratio is a brief and valid self-report measure that can be used to predict actual exercise behavior (Garcia & Archer, 2014c). Moreover, measures of personality should be included to fully address the question of factors that increase academic achievement. For instance, Moreira et al. (2012) have shown that Persistence, a temperament trait in Cloninger’s psychobiological model of personality, is a significant predictor of grade point avaarage and related to intelligence as measured by Cattell’s Reasoning Scale (see also Mousavi et al., 2014).

Figure 1 A dual focus approach simultaneously influencing well-being and academic achievement.

Implications and conclusions

According to the Swedish national agency for education (SNAE, 2009), Swedish pupils’ grades are lower than ever. According to the report by the Programme for International Student Assessment (PISA; SNAE, 2012), none of the other 32 OECD countries has fallen so far behind in grades between the years 2009–2012. For example, Natural Science scores in the PISA tests have dropped from 495 to 485 points, while the international average among the OECD contries is 501 points. Only six other countries have fewer points (SNAE, 2012). At the same time, researchers are concerned with the decay in physical education hours among Swedish high school pupils (Sverigesradio, 2013). In this context, our findings suggested that a dual (in)direct approach might increase pupils’ academic achievement and well-being—assessment being directly related to higher academic achievement, while locomotion being mainly related to frequent-intensive exercise behavior and higher well-being, both of these in turn increasing academic achievement (see Fig. 1).

“Education is the most powerful weapon we can use to change the world”

Nelson Mandela

We would like to direct our gratitude towards Åse Andrén-Gustavsson, Tommy Pettersson, and Sven-Olof Lundkvist for their openness and helpfulness in allowing the data collection.

Additional Information and Declarations

Competing Interests

Author Contributions

Human Ethics

Data Deposition

The authors declare there are no competing interests.

Danilo Garcia conceived and designed the experiments, performed the experiments, analyzed the data, wrote the paper, prepared figures and/or tables, reviewed drafts of the paper.

Alexander Jimmefors performed the experiments, analyzed the data, wrote the paper, reviewed drafts of the paper.

Fariba Mousavi, Lillemor Adrianson and Patricia Rosenberg reviewed drafts of the paper.

Trevor Archer conceived and designed the experiments, wrote the paper, prepared figures and/or tables, reviewed drafts of the paper.

The following information was supplied relating to ethical approvals (i.e., approving body and any reference numbers):

The research protocol (e.g., confidentiality, participants’ data will not be used for commercial or other non-scientific purposes) was approved by the University of Gothenburg and required only informed consent from participants.

The following information was supplied regarding the deposition of related data:

Researchgate: https://www.researchgate.net/publication/274252481_Data_Self-regulatory_mode_well-being_and_exercise_behavior.

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
