# Peer review of "Self-regulatory mode (locomotion and assessment), well-being (subjective and psychological), and exercise behavior (frequency and intensity) in relation to high school pupils’ academic achievement"

_PeerJ, doi:10.7717/peerj.847_

## Round 0.1 · original submission · Major Revisions

Dear Authors,

In your rebuttal, please seriously consider the revisions suggested by the three peer reviewers who will re-review the manuscript after re-submission.

Thank you.

·

Basic reporting

1. The first sentence of the background section of the abstract is not appropriate. At first is seems to be a general sentence about self-regulation, but then the sentence finishes with “assessment and locomotion”. I do recommend remove this sentence.
2. If possible, add some information about the statistical procedure in the abstract.
3. “Assessment” in the results section of the abstract should read “academic assessment”.
4. The conclusion of the abstract is confusing. Describe only the main conclusion and do not propose an approach.
5. Lines 19-21. “It is then important that individuals who work with any form of education, for example, teachers and other personnel in the educational system have knowledge of the different factors that affect students’ grades.” Education is more than the students’ marks. Therefore, this sentence has to be rephrased; otherwise it seems that for the authors education is reduced to students’ mark.
6. Lines 21-23. The purpose of the study should be moved to the end of the introduction. If authors decide to keep the purpose where it is, then the previous text does not provide enough rationale to sustain the purpose of the study.
7. The rationale is sometimes confusing. Try to focus only in the results of young people studies. If possible, the authors should condense the information about self-regulation (lines 42-84).
8. I recommend removing the hypothesis. When studies have hypothesis there is a problem related with self-fulfilling prophecy. Most studies published in scientific journals do not have hypothesis. Therefore, the goal of the study has to be clear, and I do recommend putting it at the end of the introduction (it is only a suggestion).
9. There are many mistakes in the references. Authors have to use the same norm for all references.

Experimental design

I have some reservations related to the experimental design that I am going to address:
1. Lines 148-150. There is no ethical statement. The authors only mentioned that it is compulsory, by law, to have informed consent from participants, but did not mention whether they get this informed consent or not.
2. Line 158. “because of legal complications with pupils under this age” it is not a reasonable motive to select a sample in a scientific study. The selected sample has to be related to be purpose of the study.
3. What was the response ratio?
4. Lines 165-166. “were sent to the authors of this article in an Excel file” Should read “were sent to the research team”.
5. For all the constructs from the different used scales there was no information of the reliability statistics (Cronbach's Alpha). In fact this information appears above, however, I recommend to put this near by the explanation of the constructs.
6. Line 211. I am not familiar with “The Archer Ratio” and I am afraid that most readers are not too. So, provide more information about this measure of exercise behaviors.
7. Line 227. The parenthesis in the middle of the line has to be removed.

Validity of the findings

1. Line 249. “while locomotion was weakly associated”. Based on the study results there was not a significant relationship between assessment and locomotion. Thus, this information is irrelevant.
2. Lines 249-250. I recommend removing “(below .20 which is the minimum effect size presenting a "practically" significant effect for social science data).” If authors want to keep this sentence they have to provide a reference to support.
3. Lines 250-258. The sentence was based on the observed relationship between assessment and grade point average (outcome variable). So, it is not clear what the authors want to prove when trying to correlate assessment with self-regulation. Instead of clarifying the readers the text is rather confusing.
4. Lines 264-265. Authors should not ask the readers to see other paper “For similar results regarding emotional well-being see Batenburg-Eddes and Jolles’ study (2013).”
5. Line 286. Instead of Johnson (2008) use another reference published in scientific journals
6. Line 269. What does “agency, for instance, is part of a mature character” means?

Additional comments

I believe the paper has potential to be published. However, needs several improvements. The different variables have to be presented but interrelated at the same time. The rationale needs to be fluid and for self-regulation the authors should mention the most known authors to show that they know the literature. Be careful with hypothesis. Normally, studies do not have hypothesis. The purpose is enough. All instruments have to be mentioned and the validation process and the reliability of the data for the study sample. Results are well presented. The discussion has to be focused on articles findings and not on speculation.

·

Basic reporting

Title: Self-regulation (locomotion and assessment), well-being (subjective and psychological), and exercise behavior (frequency and intensity) in relation to high school pupils’ academic achievement

September 22, 2014

This paper focused on the associations between self-regulation, well-being and exercise behavior, looking at their possible interactions with academic achievement. It is an old concern from exercise and sport researchers, but its importance seems heightened given the sedentary lifestyle that most populations now have.
The paper has some merits, like its innovative approach to the problem, using the self-regulation rationale, but in my opinion still is in need of several major revisions.
Some examples are presented below:


Abstract:
Minor revisions:
I suggest that in the background you state that self-regulation can be (you state that it is) operationalised of two inter-related strategies. The Higgins perspective is one of the ways to operacionalize self-regulation, but it not the only one (see Gollwitzer work on intentions for example)

Introduction:
Minor revisions:
Line 16, you have "el al.”, it should be "et al.”

Major revisions
I suggest that the section self-regulation is the first one of the introduction: it is the key variable of your study.

Major revisions
On line 26 you include a reference from Garcia, Nima & Kjell (2014) to support the the two types of well-being. I do not believe that this reference is to be included as the one that represents this classification; I agree that the Diener (1994) is THE reference to be included here - your reference does not represent the seminal author of this idea. Please remove it from this line.

Major revisions
On line 31 you have again a reference (Garcia, 2011) from a PhD thesis that does not seem to be the most appropriate to support a literature review on this subject. This happens again on line 40. Avoid self-referencing.

Minor revisions
Sometimes you use he/she (line 44) others she/he (line 57). I believe that it should be she/he, according to the APA manual of publication.

Minor revisions
Line 63 - capitalisation of the variables name, you have “assessment” in this line and then you have “locomotion in the next line”. Please correct to give consistency to the capitalisation use.

Minor revisions
Line 79 you have et al (without the period after al) on the Kruglanski reference and on the next one (Jimmefors) you already have the period. Again please correct for consistency.

Major revisions
The exercise section should be updated regarding the references that were used and also to improve the terminology used (for example good exercise habits is somewhat vague). I suggest you look at the following works to revise this section (doi:10.1007/s10648-007-9057-0 or the report fro Trost, 2009 available on http://activelivingresearch.org/active-education-physical-education-physical-activity-and-academic-performance)

Experimental design

Methods

Please revise the text - there are a lot of minor typos, e.g., subjevtive (line 175) liker (line 176) and so on.

Major revisions
Ethical statement, I suggest that you include more information regarding your decision. I wasn’t able to retrieve the law you have referenced (maybe the reference is lacking from the text) and it seems strange that you have decided this only on the basis that the data won’t be used for commercial used or non-scientific purposes. What about the psychological burden of answering the questionnaires, for example.

Minor revisions
During the descriptions of the measures you only mentioned the psychometric characteristics of the Psy WB questionnaire (because it was poor). I suggest you mention the psychometric characteristics of the other questionnaires also. Also I do not know if you have used both PANAS scales or a total score - the results suggest that you use a total score

Major revisions
Statistical analysis, please clarify about which selection method (if any) was used during the multiple regression analysis (ENTER, Stepwise and so on). Also you do not state any assumptions checking (normality and so on)


Major revisions
The use of r as an estimate of effect size is discussed on the Fergusson (2009) paper. The paper states that some corrections formulas should be used (page 534). have you looked at those. It is quite uncommon to refer to r as an estimate of effect size. See also Andy Field’s book (2009) where he suggests the use of r2 instead of r.

Validity of the findings

Results

Minor revisions
Line 232 and 234, the grammar does not seem right (I’m not a native english, so I may be wrong here). Please confirm.

Minor revisions
Line 236, if assessment if negatively related to SWB shouldn’t the r be negative?

Major revisions
Please refer the variance explained by the regression on the text.

Major revisions
I suggest that you present some type of effect size indicator for the r. Also that you present a way to let us know the individual contribution of each of the predictors on the multiple regression. You can use the squared semi-partial correlation, for example (see, Cohen, Cohen, West, & Aiken, 2003). This will add up information about the relative variance explained by each of the predictors.

Major revisions
The tables do not seem to be following a publication norm. Please correct this (although I haven’t found any norms for the PeerJ, I suggest you follow, for example, the APA manual).


Major revision
Why don’t you have a Cronbach’s alpha for the SWB (did you use a total score of the PANAS?)

Discussion
Major revisions:
In my opinion the discussion is correct (although I suggest some proof-reading). I think that you should not only state the limitations of the study, but also some ideas about how to overcome them. For example the Archer Ratio seem to be a limitation to measure exercise behavior; averaging the grades with the inclusion of the PE grade might create a bias... but other should be pointed out.

Additional comments

Please see my comments above.

·

Basic reporting

I commended the authors for taking the initiative to conduct this study. This type of study is needed for researchers, in particular to the Swedish population. Although the number of exercise adherence related studies conducted by the Swedish universities is growing, this type of research is surprisingly few. Overall, the manuscript need more seriousness in formatting, use of language, and the style of writing (need to adhere to APA style 6th edition). Please take caution on the English style, and stick with it, rather than switching between American English and British English. For example, in the abstract, the “Behavior” was used in the first two paragraphs, but “behaviour” was used in the third paragraph. For reference, need to check carefully.

Experimental design

The experimental design was adequate and fairly well-written. However, in line 148, need to state whether this research is approved by any ethical committee board. Statistic - line 227, suggested stating the "Result of pearson correlation" recommended by....

Validity of the findings

No comments. Excellent.

Additional comments

Overall, this manuscript needs minor corrections, as suggested below:
Abstract - strategies: Assessment
Moreover, replaced by Besides.
Assessment and locomotion scales to measure....
Psychological well-being scales
Temporal satisfaction with life scale
Positive affect and negative affect schedule

Line 16, standard replaced to quality
Line 17, use e.g., rather than i.e.,
Line 19, delete then
Line 21, The present study investigate
Line 26, reference – Garcia, Nima, & Kjell (APA)
Line 27, an affective component
Line, 30, his or her
Line, 36, purpose in life, and general growth
Line 46, suggested by Kruglanski et al. (2000)
Line 49, deleted “and” (2) Locomotion
Line 54, striving “to achieve the goal”
Line 57, he or she
Line 58, assess difference
Line 58, his or her
Line 58, behaviour
Line 60, evaluate achievement goal factprs briefly
Line 65, positively correlated…. Negatively correlated
Line 87, Lotan, Merrick, & Carmeli, 2005a; 2005b)
Line 88, e.g., rather than ie.
Line 89, positive and healthy lifestyle
Line 90, Archer & Garcia, 2014a; 2014b)
Line 92, Norris, Carroll, & Cochrane, 1992
Line 94, Blizzard, Lazarus, & Dean, 2001
Line 104, Pintrich & De Groot, 1992) same for line 121, and 257
Line 105, 106, 142, 255 e.g.,
Line 119, Bandura, 1977; 1982
Line 122, Moradi, & Andersson-A……
Line 140, Webber, & Berenson, 1996
Line 153, SD rather than ds
Line 175, Watson, Clark & Tellegen, 1988
From line 168-220, questionnaires: Need to cite the power of each Swedish version’s questionnaire used, and whether each subscales is acceptable.
Line 193, Ryff, & Wheaton, 2001
Line 204, Rosenberg, & Archer, 2014
Line 211, the archer ratio
Line 227, Delete Cronbach’s alpha, and replaced by “Results of Pearson correlation….”
Line 241, delete “For more details, see table 1”
Line 241, Further analysis using multiple linear regression showed that
Line 241, don’t use p<0.08, either p<0.001, or p = 0.008, similar for line 242
Line 253, Kruglanski et al., 2000
Line 260 Perry et al., 1990
Line 264, see Batenburg Eddes and Jolles’s study….. redundancy, please state see what….
Line 267, consisted of several
Line 276, Do not start a sentence with Suggesting that (APA style)
Line 279, 290 suggested
Line 284, Carlson et al., 2008
Line 303, suggested that
Line 313 Archer, T., & Garcia, D. (2014a). Physical
Line 316, Archer, T., & Garcia, D. (2014b).
Line 322, Psychological Review, 84(2), 191-215
Line 330, The Sciences and Engineering
Line 331 Koh, H.W., & Dietz, W. H.
Line 344, Diener, E. (1984).
Line 350, Garcia, D. (2011).
Line 358, Andersson-Arnten, A. C. (2012).
Line 368, Garcia, D., Nima, A. A., & Kjell, O. (2014).
Line 373: Doi:…… unfinish
Line 380: Journal of Economic Psychology, 38, 80-89.
Line 385, NY. Delete New York. Either one.
Line 426 The happiness-increasing strategies scales in a sample of Swedish adolescents.
Line 432 The temporal satisfiaction with life scale.
Line 436 66(2), 459-465
Line 451, Schimmack, U. (2007).
Line 454, Archer, T., & Garcia, D. (2013).
Line 457, Skolverket, ?? (2009). Same for 460, 462, 464.
Line 468, Twisk, J. W. (2001)……: A critical review

---

## Round 0.2 · accepted · Accept

Dear Authors,Thank you for submitting your revised manuscript.This manuscript has been accepted and will undergo routine processing to prepare it for publication.

·

Basic reporting

After authors revision, this paper is recommended for publication.

Experimental design

OK. Corrections were made.

Validity of the findings

OK. Corrections were made.

Additional comments

This paper is recommended for publication.